# Drug-resistant focal epilepsy in children is associated with increased modal controllability of the whole brain and epileptogenic regions

Aswin Chari [1,2], Kiran K. Seunarine[1,2], Xiaosong He[3,4], Martin M. Tisdall[1,2], Christopher A. Clark[1], Dani S. Bassett [3,5], Rod C. Scott [1,6,7,8,10 ✉] & Richard E. Rosch [3,6,9,10]

Network control theory provides a framework by which neurophysiological dynamics of the brain can be modelled as a function of the structural connectome constructed from diffusion MRI. Average controllability describes the ability of a region to drive the brain to easy-to-reach neurophysiological states whilst modal controllability describes the ability of a region to drive the brain to difficult-to-reach states. In this study, we identify increases in mean average and modal controllability in children with drug-resistant epilepsy compared to healthy controls. Using simulations, we purport that these changes may be a result of increased thalamocortical connectivity. At the node level, we demonstrate decreased modal controllability in the thalamus and posterior cingulate regions. In those undergoing resective surgery, we also demonstrate increased modal controllability of the resected parcels, a finding specific to patients who were rendered seizure free following surgery. Changes in controllability are a manifestation of brain network dysfunction in epilepsy and may be a useful construct to understand the pathophysiology of this archetypical network disease. Understanding the mechanisms underlying these controllability changes may also facilitate the design of network-focussed interventions that seek to normalise network structure and function.

[1] Developmental Neurosciences, Great Ormond Street Institute of Child Health, University College London, London, UK. [2] Department of Neurosurgery, Great Ormond Street Hospital, London, UK. [3] Department of Bioengineering, Electrical & Systems Engineering, Physics & Astronomy, Neurology, and Psychiatry, University of Pennsylvania, Philadelphia, PA, USA. [4] Department of Psychology, School of Humanities and Social Sciences, University of Science and Technology of China, Hefei, Anhui, China. [5] Santa Fe Institute, Santa Fe, NM, USA. [6] Department of Paediatric Neurology, Great Ormond Street Hospital, London, UK. [7] Department of Neurological Sciences, University of Vermont, Burlington, VT, USA. [8] Department of Paediatric Neurology, Nemours Children's Hospital, Wilmington, DE, USA. [9] MRC Centre for Neurodevelopmental Disorders, King's College London, London, UK. [10]These authors jointly supervised this work: Rod C. Scott, Richard E. Rosch. ✉email: rodney.scott@nemours.org

D isruptions to structural and functional brain networks are systems-level mechanisms of many neurological and psychiatric disorders. Structural networks are commonly assessed using diffusion-weighted magnetic resonance imaging whereas functional networks can be assessed with functional magnetic resonance imaging, electroencephalography (EEG), magnetoencephalography or intracranial electroencephalography (iEEG) techniques, among others. Epilepsy is one of the common neurological disorders associated with measurable network dysfunction. A variety of different modalities have been used to examine mechanisms of seizure dynamics and associated comorbidities, investigate the effects of treatment and, increasingly, guide the development of optimal management strategies[1–12]. Surgical treatments have been shown to alter whole-brain network dynamics, but the implications of these changes for ongoing seizure risk and cognitive outcomes are yet to be fully elucidated[10,13].

In children with epilepsy, many of whom have concurrent cognitive and psychological comorbidities, the acquisition of functional network data, particularly in the awake state, may be difficult. In light of recent advances in systems engineering, the use of structural brain networks to simulate features of neurophysiological dynamics is an attractive alternative[4,11,12]. More specifically, the concept of network control theory has been applied to the study of brain networks, describing how the structure of a networked system such as the brain, both supports and constrains functional dynamics[14,15]. It provides an understanding of the networked system's response to internal or external perturbation[14–16]. As applied to the brain, control theory metrics can be calculated from structural connectivity matrices derived from in-vivo diffusion MRI, after assuming a particular form for the network's dynamics[15]. The nodes are anatomically defined parcels of grey matter and the weight of the (i, j)-th edge in this undirected network is the number of white matter streamlines estimated between the parcel i and parcel j.

For the purposes of the control theory framework, we consider a 'brain state' to be a pattern of parcel activity; a single state is the magnitude of inferred neurophysiological activity across brain regions at a single time point. The temporal evolution of brain states can be modelled as a process of dynamics on the structural network. Aspects of this relationship between brain structure and dynamics can be captured by two controllability metrics: average controllability and modal controllability (Fig. 1)[14,15]. Average controllability models the ability of a region to drive the brain to easy-to-reach states; in humans, regions of high average controllability tend to be densely connected, and are involved in maintaining smooth and frequent transitions between related brain states, and are enriched in the default mode network. Modal controllability models the ability of a brain region to drive the system to a difficult-to-reach state; in humans, regions of high modal controllability tend to be weakly connected areas, allow switching between brain states that have high energy barriers between them, and are enriched in frontoparietal and cingulo-opercular control systems[14,15]. In brain networks, an inverse relationship between these two measures of regional controllability has been reported, which is not necessarily present in other network types[15,17]. This fact suggests parcels that can heavily modulate easy-to-reach states have a limited impact on the modulation of difficult-to-reach states and vice-versa. It is likely that this balance plays an important role in supporting and constraining brain dynamics. Originally derived from theoretical descriptions of linear dynamics in networks, the definitions of average and modal controllability have since been successfully applied to model responses to electrical stimulation in practice[18,19].

Disruption of these structural constraints of brain dynamics could be mechanisms driving outcomes in people with epilepsy[20,21]. Existing reports of controllability in epilepsy are scarce. Bernhardt

et al. demonstrated lower average controllability in the affected hemispheres of patients with surgically treated temporal lobe epilepsy, potentially accounting for the cognitive deficits seen in these patients[22]. In complementary studies, Scheid et al. used a time-evolving controllability framework and Taylor et al. used an optimum control energy framework to guide the optimal location and minimal energy inputs for neuromodulation treatments[23,24]. Here, we sought to understand changes in controllability in a cohort of children with surgically treated drug-resistant epilepsy at a single centre. Given the propensity of the epileptic brain to enter difficult-to-reach states of pathological neurophysiological activity that manifest as seizures, we hypothesised that children with focal drug-resistant epilepsy would have higher modal controllability than healthy controls and that those with multifocal epilepsies would have even higher modal controllability than those with the unifocal disease. Our second aim was to develop an understanding of how changes in controllability arise at the network structure level. We investigated brain network organisation in patients and healthy controls by studying the relationship between average and modal controllability, hypothesising that there would be a weakening of this relationship resulting in a disorganisation of the highly specialised brain network architecture in patients with drug-resistant epilepsy. Importantly, we sought to understand whether these changes were explained by changes in established graph metrics and sought to recreate these changes in the structural connectivity matrices through simulations. Our final aim was to study changes in controllability at the regional level in individual parcels. For patients undergoing resective surgery, we hypothesised that the modal controllability of the putative epileptogenic zone that underwent resection may be higher than corresponding areas in the controls.

In testing and validating these hypotheses, our study adds weight to the concept of network-focussed interventions in drug-resistant epilepsy. By linking network topology and dynamics, controllability allows us to link the pathophysiology of epilepsy and the possible treatments in one unified framework. In this study, the framework has been applied to surgical treatments which modify the network topology but potentially apply to neuromodulatory treatments too, in which the network topology is unaltered but is harnessed to guide optimal location and stimulation parameters. Ultimately, controllability is attractive as it allows the fusion of multimodal structural and functional network data under the same framework.

## Results

**Demographics**. A total of 16 healthy controls, 52 patients undergoing resective surgery for presumed unifocal epilepsy, and 27 patients undergoing insertion of vagal nerve stimulator (VNS) for presumed multifocal epilepsy were included (age range 3–20, see Supplementary Table 1). All patients had drug-resistant epilepsy[25] and had been discussed at the epilepsy multi-disciplinary team meeting at Great Ormond Street Hospital for Children. Subjects in the resective surgery group had a mixture of temporal and extratemporal resections and were classed as seizure free (Engel Class I, 33/52) or not seizure free (Engel Classes II–IV, 19/52) at the last follow-up (median 1.5 years, range 0.6–4.1 years). Subjects in the VNS group were classed as responders (≥50% reduction in seizures, 11/27) and non-responders (<50% reduction in seizures, 16/27) at the last follow-up (median 1.9 years, range 0.9–3.0 years). The distributions of age (Fig. 2a, Kruskal–Wallis test, $p < 0.0001$) and baseline cognitive function measured by the pre-operative full-scale IQ (Fig. 2b, available in 86/95 subjects, Chi-Sq test, $p = 0.01$) were not matched between the groups and were accordingly used as a covariate in subsequent analyses.

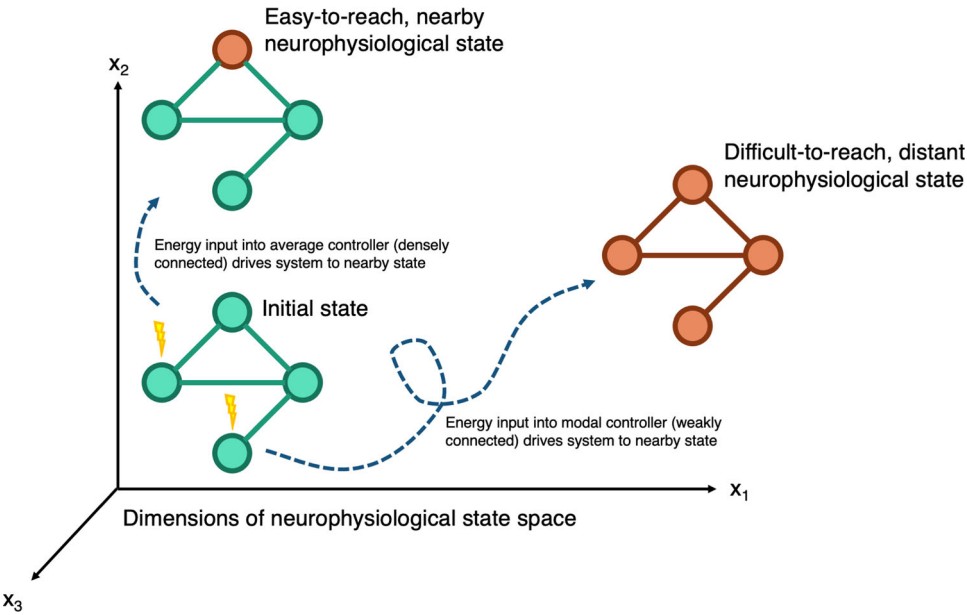

**Fig. 1 Schematic of network control theory, illustrating concepts of average and modal controllability.** Brain networks exist in a neurophysiological state space, where the state is defined as the inferred magnitude of neurophysiological activity at a given time. Nodes with high average controllability tend to be densely connected and drive the system to nearby, easy-to-reach neurophysiological states. Nodes with high-modal controllability tend to be weakly connected and drive the system to distant, difficult-to-reach neurophysiological states.

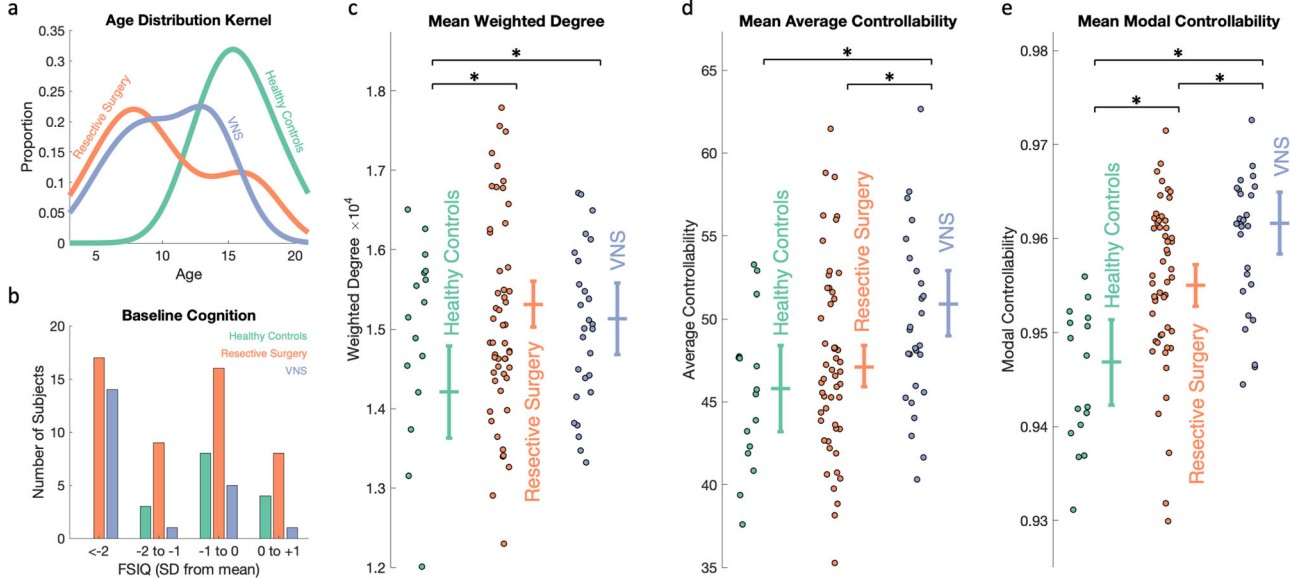

**Fig. 2 Drug-resistant epilepsy is associated with higher mean weighted degree, average controllability and modal controllability compared to healthy controls. a** Kernel plot of age distributions which were significantly different between the groups (Kruskal–Wallis test, $p < 0.0001$). **b** Baseline cognitive function measured by full-scale IQ (FSIQ) classes compared to age-corrected normal population. More epilepsy patients (resective surgery and VNS) were in the lower classes (Chi-Sq Test, $p = 0.01$). **c–e** Scatter plots of **c** mean weighted degree, **d** mean average controllability and **e** mean modal controllability for each group. Besides the scatter, plots are estimated marginal means (±95% Wald confidence intervals) following correction for age, sex, cognitive function and, for the controllability metrics, mean weighted degree. Multiple comparisons were adjusted for by the least significant difference. Both groups of epilepsy patients had a higher weighted degree compared to controls ($p = 0.001$ for resective surgery; $p = 0.02$ for VNS groups). The VNS group had significantly higher mean average controllability compared to both the controls ($p = 0.003$) and the resective surgery ($p = 0.001$) group. Both groups had higher mean modal controllability compared to controls ($p = 0.003$ for resective surgery; $p = 1 \times 10^{-6}$ for VNS) and the VNS group was even significantly higher than the resective surgery group ($p = 0.001$).

**Patients with epilepsy have a higher mean weighted degree, average controllability and modal controllability than healthy controls.** To test whether there were group differences in network controllability at the whole-brain level, we constructed structural connectivity matrices from pre-operative imaging derived from

the number of streamlines connecting each of 253 cortical and subcortical parcels for each subject (see Methods for details). Weighted degree (defined as the sum of weights of edges connected to the node), average controllability and modal controllability averaged across all parcels for each subject were compared

between the groups using a generalised linear model (GLM) approach, correcting for age, sex, cognition and, for the controllability metrics, mean weighted degree and adjusted for multiple comparisons using Fisher's least significant difference method (see Methods for details). Both groups of patients with epilepsy had a higher mean weighted degree compared to controls (mean difference 1101, cohen's $d = 1.0$, $p = 0.001$ for resective surgery; mean difference 918, cohen's $d = 0.8$, $p = 0.02$ for VNS) (Fig. 2c). The VNS group had a higher mean average controllability compared to controls (mean difference 5.13, cohen's $d = 1.0$, $p = 0.003$) and resective surgery patients (mean difference 3.82, cohen's $d = 0.7$, $p = 0.001$) (Fig. 2d). Both groups had higher mean modal controllability compared to controls (mean difference 0.008, cohen's $d = 1.0$, $p = 0.003$ for resective surgery; mean difference 0.015, cohen's $d = 1.8$, $p = 1 \times 10^{-6}$ for VNS) and the VNS group had a higher mean modal controllability than the resective surgery group (mean difference 0.007, cohen's $d = 0.8$, $p = 0.001$) (Fig. 2e).

These findings confirmed our hypothesis that children with drug-resistant epilepsy had higher modal controllability compared to healthy controls, indicating an increased propensity to reach distant states such as those associated with seizures. In agreement with previous studies, this whole-brain increase in modal controllability was associated with higher whole-brain average controllability[26].

As sensitivity analyses, we repeated these tests (i) without adjusting for covariates and (ii) limiting the comparisons to all subjects aged >12 to match the age distribution of the control cohort ($n = 44$) (Supplementary Note 1). Given the largely consistent findings both in terms of effect sizes and statistical significance across both sensitivity analyses, all further analyses using the entire dataset of 95 subjects.

**The node-level relationships between weighted degree, average controllability and modal controllability are less strong in children with drug-resistant epilepsy**. To understand the regional variation underlying the differences in average and modal controllability, we examined the relationships between weighted degree, average controllability and modal controllability at the level of the individual parcel. Previous studies have demonstrated that although the relationship between average and modal controllability can be negative, positive, or non-significant at whole-network level, at the node-level there is a direct relationship between weighted degree and average controllability, and an inverse relationship between weighted degree and modal controllability[15]. We hypothesised that patients with epilepsy would have weaker relationships between weighted degree, average controllability and modal controllability, indicating a loss of the network architecture that permits the specialisation of hubs for either average or modal control[17].

To investigate these relationships, we converted the heavy-tailed raw weighted degree and controllability values to ranks in each patient (Supplementary Fig. 1). Next, we calculated correlation coefficients to assess the strength of the relationship between the ranks of weighted degree and average controllability (WD-AC correlation), ranks of weighted degree and modal controllability (WD-MC correlation) and ranks of average and modal controllability (AC-MC correlation). Correlation coefficients were calculated for each subject and compared between groups correcting for age, sex, cognitive function and mean weighted degree. The WD-AC correlation coefficients were not different between the two groups of patients and controls (mean difference 0.04, cohen's $d = 0.4$, $p = 0.31$ for resective surgery and mean difference 0.08, cohen's $d = 0.6$, $p = 0.10$ for VNS). There was a significant difference in WD-MC correlation coefficients between the two groups and controls (mean difference 0.03,

### Rank of average controllability vs rank of modal controllability

**a. Healthy Controls**

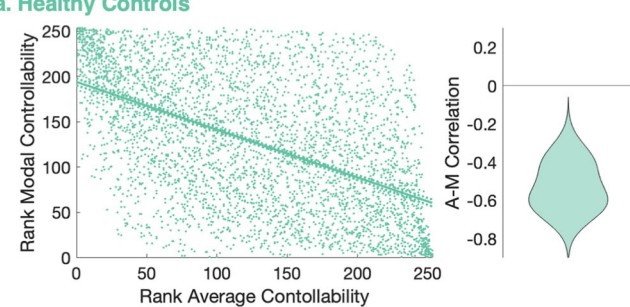

**b. Resective Surgery**

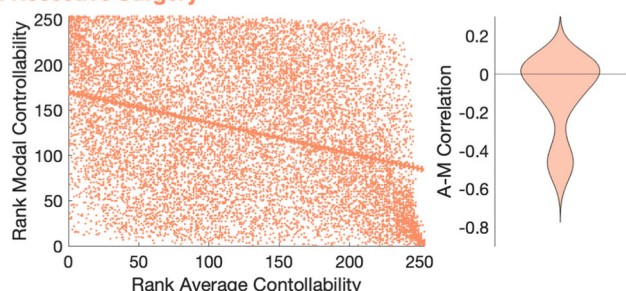

**c. VNS**

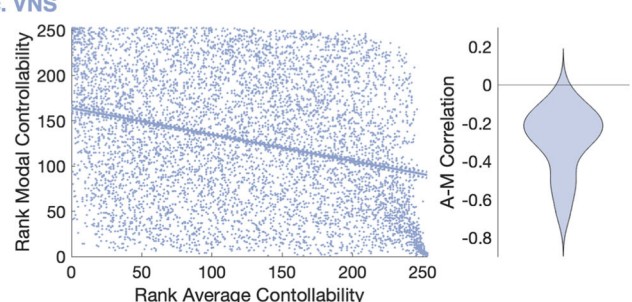

**Fig. 3 The inverse correlation between average controllability and modal controllability is weaker in patients with drug-resistant epilepsy.** **a–c** Correlation between the rank of average controllability and rank of modal controllability (AC-MC correlation) for each parcel across all **a** healthy controls, **b** resective surgery and **c** VNS patients. Lines show the linear regression best fit (solid line) and 95% confidence intervals (dotted lines). Adjacent violin plots show distribution of these correlation coefficients for each individual subject. There were significant differences in the distributions of the correlations at the individual patient level between controls and both groups of patients ($p = 7 \times 10^{-8}$ for resective surgery and $p = 0.02$ for VNS after correcting for age, sex, cognitive function, mean weighted degree and multiple comparisons).

cohen's $d = 0.8$, $p = 0.02$ for resective surgery and mean difference 0.04, cohen's $d = 1.2$, $p = 0.001$ for VNS). Both groups also had a significantly weaker AC-MC correlation compared to the healthy controls (mean difference 0.34, cohen's $d = 1.7$, $p = 7 \times 10^{-8}$ for resective surgery and mean difference 0.15, cohen's $d = 0.8$, $p = 0.02$ for VNS; see Fig. 3). Together, these results indicate an alteration of the structural brain networks in patients with epilepsy that affects the separation of hubs with high average controllability and those with high modal controllability into non-overlapping brain areas.

**The controllability changes are not explained by graph measures but are explained by additional thalamocortical connections in the epileptic brain**. In order to assess whether the network-level

changes in controllability were merely reflective of changes in existing graph metrics that have been associated with drug-resistant epilepsy, we compared three measures, namely modularity, global efficiency and diffusion efficiency (see Methods for definitions) between groups[27]. Compared to healthy controls, both groups of patients had a lower modularity ($p = 9 \times 10^{-6}$ for resective surgery and $p = 4 \times 10^{-6}$ for VNS) and higher diffusion efficiency (cohen's $d = 1.6$, $p = 5 \times 10^{-7}$ for resective surgery and cohen's $d = 1.8$, $p = 5 \times 10^{-7}$ for VNS) but no difference in global efficiency (cohen's $d = 0.4$, $p = 0.27$ for resective surgery and cohen's $d = 0$, $p = 0.92$ for VNS) (all metrics adjusted for age, sex, cognitive function and mean weighted degree and $p$ values were corrected for multiple comparisons). To gather a conceptual appreciation of the relationship between controllability metrics and the graph metrics, we plotted the correlation between the metrics (Supplementary Fig. 2). On visual inspection, the groups were segregated most clearly by the AC-MC correlation, and this segregation did not track strongly with any of the graph metrics.

One of the keys to further our understanding of network changes in epilepsy is to dissect how a healthy network can be altered into an epileptogenic one such that we can design novel therapeutics at any level of the complex systems framework to reverse these changes. We, therefore, sought to probe what the changes were in the structural connectivity matrices that resulted in these differences in controllability and node-level correlations between patients and controls. We started by assessing the density of the structural connectivity matrices (see Methods for definition). The healthy controls had an estimated marginal mean density of 0.63, compared to 0.71 in the resective surgery group ($p = 3 \times 10^{-6}$) and 0.69 in the VNS group ($p = 0.001$) after correcting for age, sex, cognitive function and mean weighted degree and multiple comparisons. Kernel density distributions of edge weights showed minimal differences between the groups (Supplementary Fig. 3). Together with the increase in weighted degree identified previously, these findings suggested an increased number of edges in the structural networks of children with epilepsy compared to healthy controls.

We, therefore, used numerical simulations to test whether the addition of edges at random to the healthy control connectivity matrix could recreate the specific network-wide alterations observed in our patient groups. To assess the impact on the network, we used the graph and controllability metrics as a 'network fingerprint'. Adding edges at random led to an increase in average controllability but no significant change in modal controllability or the WD-AC, WD-MC and AC-MC correlation coefficients, inconsistent with our biological findings (Supplementary Fig. 4).

We, therefore, hypothesised that the 'network fingerprint' observed in our data may be due to a more anatomically constrained addition of edges. To investigate this further, we assessed the differences in mean connectivity matrices between controls and both groups of patients, identifying an increase in low streamline count (<100) edges, a majority of which were localised to ipsilateral thalamocortical connections (Supplementary Fig. 5). To assess whether increases in thalamocortical connections alone were sufficient to recreate the changes in 'network fingerprint' changes observed in our patient cohort, we conducted a further simulation by increasing a fixed number (~4.5% of the total edges) of these ipsilateral thalamocortical connection edge weights in control subjects. We compared them to the baseline connectomes and null models of increasing an equal number of edges by equal weight anywhere in the ipsilateral hemisphere (see Methods for details). The anatomically constrained models reproduced the 'network fingerprint' changes seen in people with epilepsy more closely than the unconstrained null models (Fig. 4). Specifically, compared to the controls, the anatomically constrained thalamocortical models had statistically significant increases in mean average controllability ($p < 1 \times 10^{-10}$), mean modal controllability ($p = 3 \times 10^{-9}$), WD-MC correlation ($p < 1 \times 10^{-10}$), AC-MC correlation ($p = 5 \times 10^{-4}$), global efficiency ($p = 9 \times 10^{-9}$), diffusion efficiency ($p < 1 \times 10^{-10}$) and a decrease in modularity ($p < 1 \times 10^{-10}$) after correcting for multiple comparisons (Benjamini–Hochberg FDR method). Interestingly, although the statistically significant findings were similar between the controls and the unconstrained null models, there was no significant difference in the AC-MC correlation ($p = 0.85$ after correcting for multiple comparisons). This observation shows that the changes, specifically the increase in AC-MC correlation coefficient, a weakening of the negative correlation, observed in our patient data were most closely recreated by the anatomically constrained models that selectively increased thalamocortical weights.

**Global controllability metrics do not strongly associate with treatment outcome.** Next, we sought to assess whether the controllability measures were markers for treatment success, both in the VNS and resective surgery groups (Supplementary Fig. 6). After correction for age, sex, cognitive function, mean weighted degree and multiple comparisons, we observed subtle differences in the metrics, with the VNS non-responders having a higher mean average controllability than the responders ($p = 0.02$) and the seizure-free patients following surgical resection having a higher mean modal controllability compared to the non-seizure-free patients ($p < 0.05$). The weighted degree and correlation metrics were not discriminatory between groups once stratified by the outcome.

**Thalamic and posterior cingulate parcels show increased weighted degree and decreased modal controllability in patients with drug-resistant epilepsy.** In order to assess whether there were common nodes of the brain (e.g., thalamic nuclei as informed by the simulations) that subserved the changes in mean average and modal controllability, we created a $Z$ score for the ranks of each parcel in each patient using the mean and standard deviation from the controls; this procedure normalised the data (Supplementary Fig. 1). Using a $Z$ score threshold of ±3.1 that served to correct for the 253 individual comparisons, there was a significantly higher weighted degree and lower modal controllability of select bilateral thalamic nuclei and posterior/isthmus cingulate cortices in both groups of patients compared to controls (Table 1, Fig. 5). This corroborated the findings of the simulations of increased thalamocortical connectivity that manifest in the control theory framework as lower modal controllability.

**Resected parcels show decreased weighted degree and increased modal controllability in patients undergoing resective surgery.** Given the decreased thalamic and posterior cingulate modal controllability at the group level, we postulated that cortical parcels could have an increased modal controllability. We specifically hypothesised that this could be localised to parcels that are part of the putative epileptogenic zone that undergo resection. To test this, we identified the resected parcels from the postoperative scans and compared the mean $Z$ scores for the resected and non-resected parcels (see Methods for details, Supplementary Fig. 7). The z scores were calculated using the mean and standard deviations of the ranks from the healthy controls; for the set of parcels in the control group, the mean z score would therefore be 0. In the resective surgery group, following correction for multiple comparisons, we found significant differences in the mean

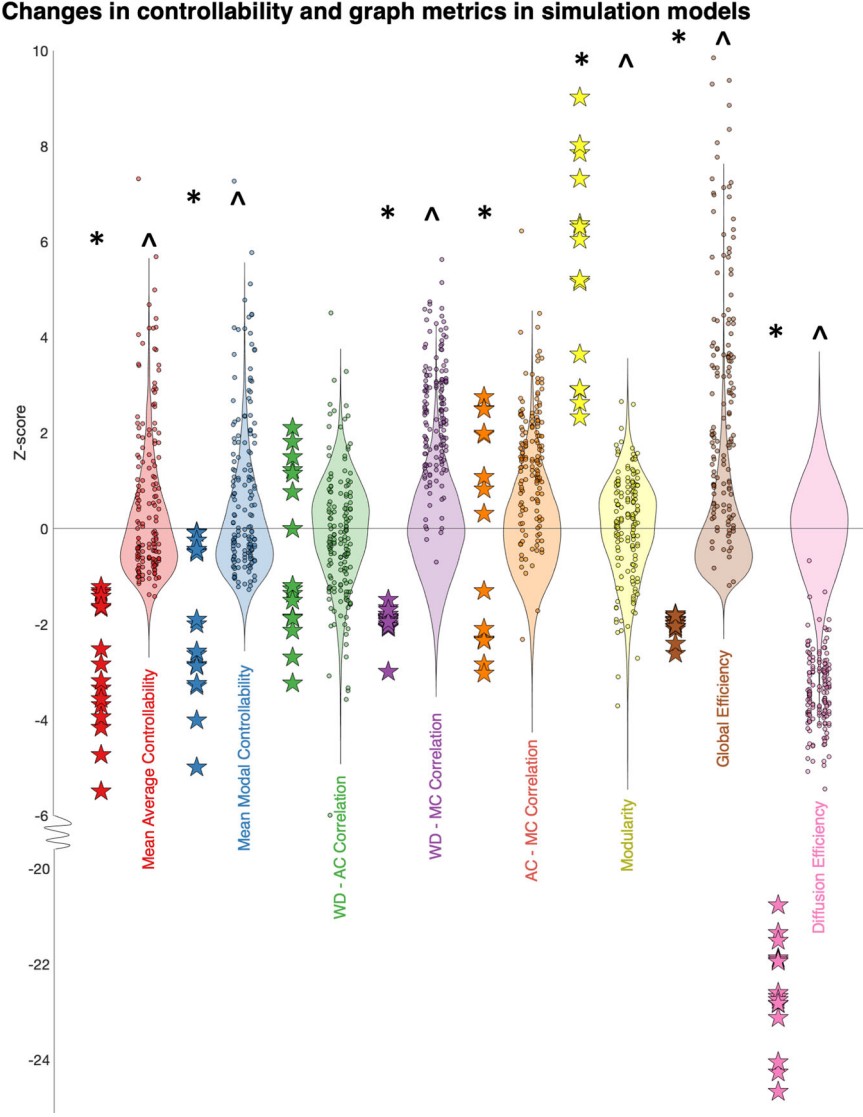

**Fig. 4 Changes in controllability and graph metrics observed in patient data are closely recreated by selectively increasing the weight of thalamocortical connections by a low number of streamlines.** $Z$ scored controllability and graph metrics are compared between controls (stars) and anatomically constrained simulations (dots, 10 for each control) of increasing ipsilateral thalamocortical edges (increase in edge weight of 4.5% (1440/31818) of edges by a streamline chosen from a normal distribution with $\mu = 50$, $\sigma = 15$). The background violin plots show the null models of increase in the same number of edges by the same weight, constrained to all ipsilateral edges only (200 for each control), on which the $Z$ scores were calculated. The distributions between the controls and constrained thalamocortical models (*) and controls and unconstrained null models (^) were compared using two-sample $t$ tests and adjusted for multiple comparisons (Benjamini–Hochberg FDR method). Compared to the controls, the anatomically constrained thalamocortical models had statistically significant increases in mean average controllability ($p < 1 \times 10^{-10}$), mean modal controllability ($p = 3 \times 10^{-9}$), WD-MC correlation ($p < 1 \times 10^{-10}$), AC-MC correlation ($p = 5 \times 10^{-4}$), global efficiency ($p = 9 \times 10^{-9}$), diffusion efficiency ($p < 1 \times 10^{-10}$) and a decrease in modularity ($p < 1 \times 10^{-10}$). Compared to the controls, the anatomically unconstrained models had statistically significant increases in mean average controllability ($p < 1 \times 10^{-10}$), mean modal controllability ($p < 1 \times 1 - 10^{-10}$), WD-MC correlation ($p < 1 \times 10^{-10}$), global efficiency ($p < 1 \times 10^{-10}$), diffusion efficiency ($p < 1 \times 10^{-10}$) and a decrease in modularity ($p < 1 \times 10^{-10}$). The specific important differentiator in our patient group (AC-MC correlation) was selectively replicated only in the anatomically constrained thalamocortical model.

weighted degree and mean modal controllability between the whole-brain and resected/non-resected parcels, only in patients that were seizure free following surgery (Fig. 6a). Seizure-free patients had a lower mean weighted degree (mean $z$ score difference 0.59, $p = 0.002$) and higher mean modal controllability (mean $z$ score difference 0.67, $p = 6 \times 10^{-4}$) of the resected parcels compared to the whole brain. A similar effect was observed when comparing the actual resections against 1000 'virtual resection' null models of other randomly selected cortical parcels (Fig. 6d, mean z score 1.12, $p = 0.002$ for weighted degree and mean z score 1.24, $p = 0.007$ for modal controllability), indicating

that this effect was specific to the clinically identified putative epileptogenic zone. There were no significant differences between the means of the resected parcels in different histological groups (Kruskal–Wallis tests, $p = 0.14$ for weighted degree, $p = 0.15$ for average controllability and $p = 0.56$ for modal controllability; groups detailed in Supplementary Data 1).

The non-resected, remaining parcels had a higher mean weighted degree (mean z score difference 0.03, $p = 6 \times 10^{-4}$) and lower mean modal controllability (mean z score difference 0.04, $p = 4 \times 10^{-4}$), again only in patients that were seizure-free following surgery (Fig. 6b). To confirm this effect, we also generated the connectivity

**Table 1 Identified parcels that had a significantly different weighted degree, average controllability and modal controllability in both groups of patients compared to healthy controls.**

| | Weighted degree > controls | Weighted degree < controls | Average controllability > controls | Average controllability < controls | Modal controllability > controls | Modal controllability < controls |
|---|---|---|---|---|---|---|
| Resective surgery | - Right pulvinar<br>- Right central-lateral, lateral-posterior, medial-pulvinar<br>- Left isthmus cingulate 1<br>- Left pulvinar<br>- Left central-lateral, lateral-posterior, medial-pulvinar | | | | | - Right posterior cingulate 2<br>- Right isthmus cingulate 1<br>- Right central-lateral, lateral-posterior, medial-pulvinar<br>- Right ventral-latero-ventral<br>- Left isthmus cingulate 1 |
| VNS | - Right-pulvinar<br>- Right ventral-latero-dorsal<br>- Right central-lateral, lateral-posterior, medial-pulvinar<br>- Left posterior cingulate<br>- Left isthmus cingulate 1<br>- Left pulvinar<br>- Left central-lateral, lateral-posterior, medial-pulvinar | - Left pericalcarine 1 | - Left lateral orbitofrontal 2<br>- Left hypothalamus | - Right lateral occipital | | - Right posterior cingulate 2<br>- Right anterior thalamus<br>- Right ventral-latero-dorsal<br>- Right central-lateral, lateral-posterior, medial-pulvinar<br>- Right ventral-latero-ventral<br>- Left posterior cingulate 1<br>- Left isthmus cingulate 1<br>- Left medio-dorsal<br>- Left ventral-latero-dorsal<br>- Left ventral-latero-ventral |

Parcels that had a significantly different (average *Z* score threshold of ±3.1) weighted degree, average controllability and modal controllability for the resective surgery and VNS cohorts.

**Parcels with lower modal controllability compared to controls**

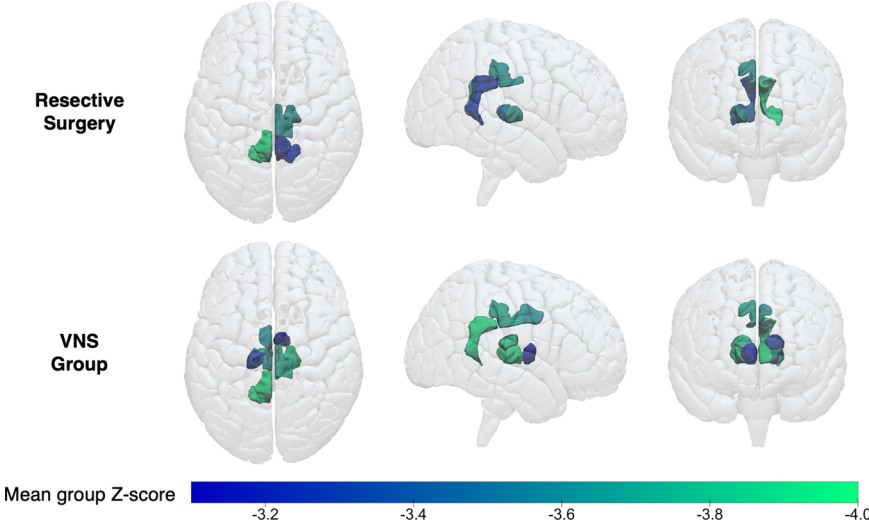

**Fig. 5 Patients with drug-resistant epilepsy show lower modal controllability of thalamic and posterior cingulate parcels.** Visual maps of the parcels with significantly lower modal controllability (mean *Z* scores < −3.1) in the resective surgery and VNS groups. Note the concentration of parcels around the thalamic nuclei and posterior cingulate cortices. The exact parcels are detailed in Table 1.

matrices again on the pre-operative scans, excluding the resection volumes from the possible streamlines (see Methods for details). We found that this expected post-operative connectome had a higher mean weighted degree and lower mean modal controllability but these findings were significantly different for patients that were both seizure-free (mean $z$ score difference 0.06, $p = 5 \times 10^{-4}$ for weighted degree and mean z score difference 0.05, $p = 3 \times 10^{-6}$ for modal controllability) and not seizure-free (mean $z$ score difference 0.09, $p = 3 \times 10^{-5}$ for weighted degree and mean z score difference 0.06, $p = 4 \times 10^{-5}$ for modal controllability) following correction for multiple comparisons. There was also an increase in

average controllability in those that were seizure free ($p = 0.001$) (Fig. 6c).

## Discussion

In this diffusion MRI study of 79 children with drug-resistant epilepsy, we identified increases in network mean weighted degree, average and modal controllability in patients compared to controls (Fig. 2). The changes in modal controllability have not previously been explored in the context of drug-resistant epilepsy. The graded increase between patients undergoing resective surgery (clinically characterised as unifocal epilepsy) and VNS

## a. Controllability metrics in the resected parcels

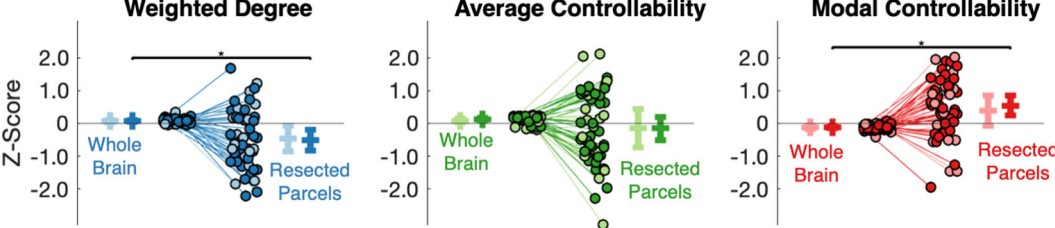

## b. Controllability metrics in the non-resected parcels

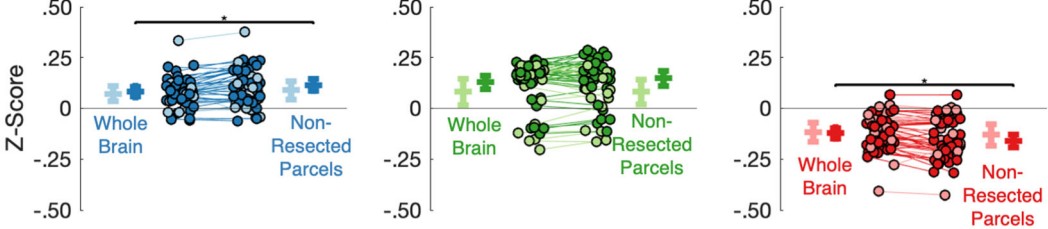

## c. Controllability metrics in the post-operative connectome

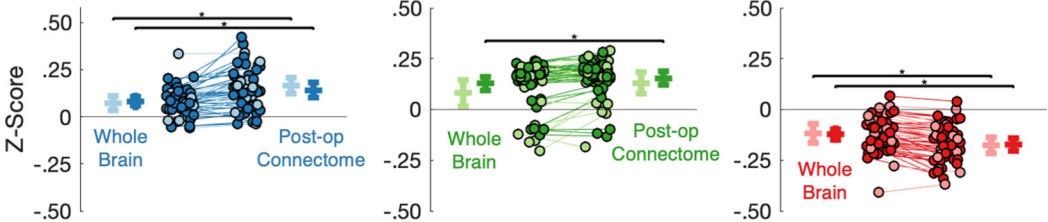

## d. Controllability metrics of the real resection compared to 1000 virtual resections

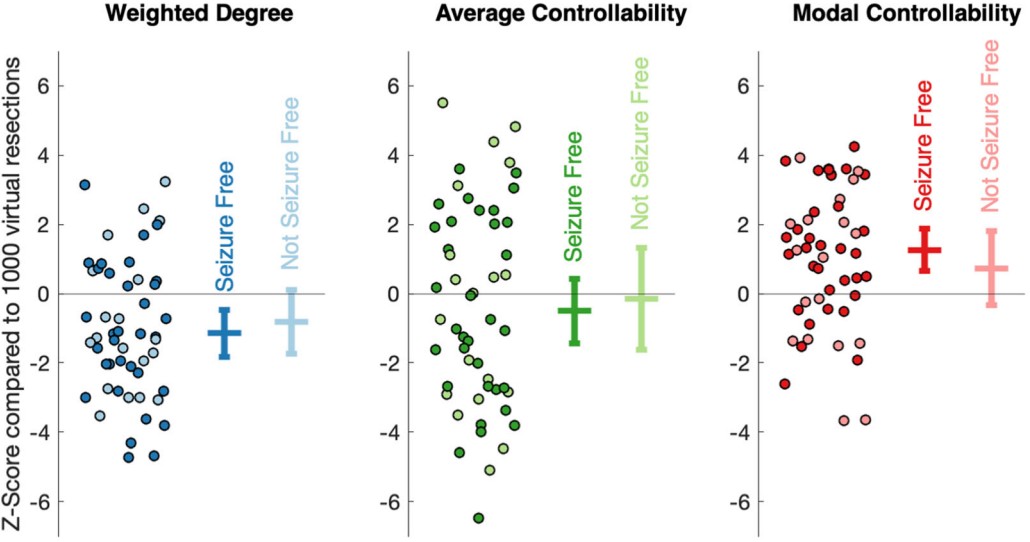

**Fig. 6 Resected parcels in seizure-free patients show a lower weighted degree and higher modal controllability than equivalent parcels in controls and compared to virtual resections in patients. a–c** Changes in the mean $Z$ scores for weighted degree, average controllability and modal controllability for each resective surgery patient between baseline, the **a** resected parcels, **b** the non-resected parcels and **c** the post-operative connectome. There were significant differences in the mean weighted degree ($p = 0.002$ and $p = 6 \times 10^{-4}$) and modal controllability ($p = 6 \times 10^{-4}$ and $p = 4 \times 10^{-4}$) for both resected and non-resected parcels in seizure-free patients only (paired $t$ tests, adjusted for multiple comparisons using the Benjamini–Hochberg FDR method). For the post-operative connectome, there were significant differences in the mean weighted degree for both seizure-free and not seizure-free patients ($p = 5 \times 10^{-4}$ and $p = 3 \times 10^{-5}$, respectively), mean average controllability in seizure-free patients only ($p = 0.001$) and mean modal controllability for both seizure-free and not seizure-free patients ($p = 3 \times 10^{-6}$ and $p = 4 \times 10^{-5}$, respectively) (paired $t$ tests, adjusted for multiple comparisons using the Benjamini–Hochberg FDR method). **d** $Z$ scores of weighted degree, average controllability and modal controllability for the real resection compared to 1000 virtual resections for each of the patients, stratified by post-operative outcome. Bars show mean ±95% confidence intervals. The real resection parcels were significantly different from 0 for weighted degree ($p = 0.007$) and modal controllability ($p = 0.002$) in seizure-free patients only (one-sample $t$ tests, adjusted for multiple comparisons using the Benjamini–Hochberg FDR method).

implantation (clinically characterised as multifocal epilepsy) even when age, cognitive effects and mean weighted degree are corrected for, suggests that there are underlying network architectural differences in children with drug-resistant epilepsy that result in the brain being more easily able to reach difficult-to-reach states of neurophysiological activity. We interpret this as the brain being more easily able to transition to a seizure state, which is defined as 'abnormal excessive or synchronous neuronal activity in the brain'[28].

We found indicators of a loss of topographic specialisation in brain networks of patients with epilepsy in terms of the role of individual brain areas in shaping whole-brain dynamics. Wu-Yan et al. suggest that a strong correlation between average and modal controllability may represent a 'specialisation' in brain areas, which have either high modal or high average controllability (but not both)[17]. In our cohort of patients with drug-resistant epilepsy, this relationship was weaker, with increased (less negative) AC-MC correlation coefficients (Fig. 3). This loss of specialisation may have therapeutic implications. The hallmarks of epilepsy are recurrent seizures and associated behavioural and cognitive comorbidities. Therefore, assessing both modal (propensity to seizures) and average (cognitive function) changes may reveal that network structural changes unify these aspects of epilepsy and provide a framework from which to optimise both via modulatory strategies.

The mechanistic simulations we undertook allowed us to probe the changes in the network structure underlying this disorganisation. By exploring differences in the structural connectivity matrices and network properties (Supplementary Figs. 2 and 4), we hypothesised that these changes may occur due to increased connectivity in the epileptic brain, specifically in thalamocortical connectivity. Selectively increasing thalamocortical connections in control subjects by a low number of streamlines recapitulated many of the network properties seen in patients with epilepsy; including increases in average and modal controllability, weakening of the AC-MC correlation coefficient, decreased modularity and increased diffusion efficiency (Fig. 4). This observation is in agreement with other studies that have found increased thalamocortical connectivity in children with drug-resistant epilepsy[29]. Elucidating the nature of these changes may be important for complex system-based approaches to epilepsy treatment such that our interventions, at any layer of the complex system hierarchy, can reverse these changes back towards that of a 'normal' healthy network[1,30].

A key question that arises from this set of findings is whether the increased thalamocortical connectivity and the resultant impacts on controllability are mechanisms for epilepsy or are a result of uncontrolled seizures in this drug-resistant population. Epilepsy is also associated with a plethora of psychological and cognitive comorbidities, and it would be prudent to assess whether these and epileptic seizures are subserved by common underlying network alterations. Large longitudinal datasets from, for example, the ENIGMA consortium should be able to answer such questions[31,32]. If changes in controllability are shown to progress with time, it may make the case for earlier surgical intervention, perhaps even before the development of drug resistance.

At the node-level, our study specifically identified decreased modal controllability in the cingulate cortex and thalamus of patients with drug-resistant epilepsy (Table 1). The cingulate cortex parcels have been shown to be part of the structural core of the cortex, with high node degrees and topographical centrality important, for functional integration[33]. The thalamic parcels correspond to the locations of the centromedian and anterior nuclei, both established targets for deep brain stimulation as a treatment for epileptic seizures; the anterior nucleus is thought to be a better target for focal epilepsy whereas the centromedian is thought to be a better target for generalised epilepsy (Supplementary Table 1)[34–36].

The involvement of multiple thalamic nuclei underscores the importance that thalamocortical circuitry plays in seizures and epilepsy[37–40]. The increased connectivity of both regions lowers modal controllability values, perhaps indicating a reduced ability of these brain regions to move the brain into difficult-to-reach neurophysiological states (e.g., out of a seizure).

The resected parcels in patients undergoing surgery and rendered seizure free had a lower mean weighted degree and higher mean modal controllability than the same parcels in controls (Fig. 6a) and compared to virtual resection in patients (Fig. 6d), suggesting an increased propensity for these parcels to reach difficult-to-reach neurophysiological states such as seizures. The finding of decreased weighted degree of these parcels complements previous studies that have noted variable connectivity patterns in focal cortical dysplasia, finding lesions with increased and decreased connectivity that corresponded to FCD type IIa and IIb, respectively[41]. In our more heterogeneous cohort, we did not identify such differences in the weighted degree or controllability metrics between histological groups. Indeed, the direction of change of modal controllability seemed more consistent than the direction of change of weighted degree (Fig. 6a), suggesting a propensity of these brain regions to move the brain into a seizure state.

At whole-brain scale, we have progressed our understanding of brain network dysfunction in drug-resistant epilepsy, showing that the structural network alterations are constrained to more easily reach nearby and distant neurophysiological states given a perturbation. This perturbation may be caused by extrinsic inputs or intrinsic local abnormalities that, for example, affect excitatory-inhibitory balances, leading to a transition into a seizure state[42]. The graph metrics and correlations illustrate that this is a function of a less organised and less modular system, which the simulations suggest may in turn be due to increased thalamocortical connectivity.

Other studies have used the control theory framework to simulate responses of the brain to external perturbation via stimulation[18,19,24]. At node level, our demonstration of regional differences in modal controllability in children with drug-resistant epilepsy raises the potential to use these metrics to guide clinical treatments. Firstly, there is the possibility of identifying parcels of abnormal controllability as surrogate markers of the epileptogenic zone. Such analyses may reveal more focal, more widespread or multifocal network abnormalities that could be useful in guiding resections and prognosticating[3]. Secondly, the network dynamics from the control theory framework could be used to choose the location and stimulation parameters for neuromodulation treatments, with the aim of normalising the network dynamics[1]; this has the potential to ameliorate both the susceptibility to seizures and some of the comorbid conditions such as cognitive and behavioural dysfunction associated with epilepsy[43].

Our study had several important technical limitations. Firstly, the cohorts were not age- or sex-matched, which may have been a particular issue given that the developmental trajectories of controllability metrics have not been fully elucidated[26]. However, the patients had increased average and modal controllability despite being younger, thus if we extrapolate from previously described age effects, the effect size for group differences would have been diluted in our cohort. We used generalised linear model approaches to try adjust for some of these factors, including age, sex and cognition. In these models, our findings of changes in controllability with age are consistent with Tang et al.'s findings that there is a stronger increase in the average controllability of whole-brain networks with age compared to modal controllability[26]. This strategy allowed us to maintain recording consistency in that all subjects were scanned on the same MRI scanner with identical protocol and post-processing, mitigating a number of otherwise confounding factors.

There were also limitations that have to do with the interpretability of the findings. As recognised by previous studies, the use of network control theory to model and modify the human brain is still relatively new[19,44,45]. Our findings that there are parcels with lower modal controllability in the thalamus and posterior cingulate regions and higher modal controllability in the resected regions are robust. The impact that these observations will have on both resective and neuromodulatory strategies for epilepsy is yet to be fully elucidated. It is foreseeable that this data could be used to inform neuromodulation targets as has recently been reported[19,23].

In this study, we have identified that paediatric drug-resistant epilepsy is associated with higher whole-brain weighted degree, average controllability and modal controllability and lower modal controllability of the thalamus and posterior cingulate parcels. In those undergoing resective surgery, modal controllability of the resected parcels is higher than would be expected and this effect is linked to post-operative outcomes. At the node level, children with drug-resistant epilepsy also have less strong correlations between weighted degree and controllability metrics, indicating a loss of topographic specialisation that may be partly due to increased thalamocortical connectivity. The findings emphasise the concept of more widespread changes being associated with 'focal' epilepsy and the importance of network-based approaches to better understand the pathophysiologic mechanisms and, in future, guide treatment[9,10].

## Methods

This retrospective study was conducted following approval from the Joint Research Office of Great Ormond Street Hospital & University College London Institute of Child Health (project ID 19BI26). As it was a retrospective study involving routinely collected clinical data, individual patient consent was waived.

**Subjects**. Subjects were eligible for inclusion if they had undergone volumetric T1 MPRAGE and multi-shell diffusion imaging between 2015 and 2019 in one of three categories.

*Resective surgery*. Patients aged >3 who had undergone an MRI scan as part of the pre-operative evaluation for resective epilepsy surgery. Surgery was carried out following discussion at a specialist epilepsy surgery multidisciplinary team meeting and some of these patients underwent intracranial evaluation with SEEG prior to proceeding with resective surgery. If there were multiple scans, the scan time-point closest to the surgery (but prior to SEEG) was chosen. To reduce the impact of large structural abnormalities on the diffusion imaging metrics, patients were excluded if they had brain tumours, tuberous sclerosis, prior neurosurgical intervention or other large structural abnormalities such as previous strokes. Resections included a combination of anteromedial temporal lobe resections (27.0%) and extratemporal resections in the form of lesionectomies and lobectomies (73.0%), the exact extent of which was driven by the pre-surgical evaluation. Post-operative histology included focal cortical dysplasia and hippocampal sclerosis (Supplementary Data 1). Only those with a post-operative volumetric scan (to identify the resection volume) were included.

*VNS*. Patients aged >3 who had undergone an MRI scan as part of the pre-operative evaluation for a VNS. All patients were also discussed at a specialist epilepsy surgery multidisciplinary team meeting and deemed unsuitable for surgical resection due to presumed multifocal aetiology of seizures. To reduce the impact of large structural abnormalities on the diffusion imaging metrics, patients were excluded if they had tuberous sclerosis, prior neurosurgical intervention or other large structural abnormalities such as previous strokes.

*Healthy controls*. Controls were scanned as part of another research study and were unaffected siblings of children with sickle cell disease. None of these children had epilepsy and the ones included had no significant MRI abnormalities on neuroradiological review[46].

Demographic information is available in Supplementary Data 1. Race and ethnicity in the groups was likely to be different but this data was not available for all subjects. However, race and ethnicity do not seem to have a major impact on grey and white matter volumes[47] and, given the internal rankings used in the latter analyses (see below), this was not considered a major confound.

The entire workflow for image processing is summarised in Fig. 7 and explained in detail below.

**MRI scanning**. All patients and controls were scanned on the same Siemens Magnetom Prisma 3.0 T MRI scanner at Great Ormond Street Hospital, equipped with a 20-channel head coil. All scans (apart from the healthy controls) were acquired for routine clinical purposes.

The protocol included a T1 MPRAGE sequence and multi-shell diffusion sequence employing a diffusion-weighted spin-echo single-shot 2D EPI acquisition, with multi-band radio frequency pulses to accelerate volume coverage along the slice direction. A multi-band factor of 2 was used to image 66 slices of 2 mm thickness with 0.2 mm slice gap. Diffusion gradients were applied over two 'shells': $b = 1000$ s/mm$^2$ and $b = 2200$ s/mm$^2$, with 60 non-collinear diffusion directions per shell in addition to 13 interleaved $b = 0$ (non-diffusion-weighted) images. Other imaging parameters were: TR = 3050 ms, TE = 60 ms, field of view = 220 mm × 220 mm, matrix size = 110 × 110, in-plane voxel resolution = 2.0 mm × 2.0 mm, GRAPPA factor 2, phase-encoding (PE) partial Fourier = 6/8. An additional $b = 0$ scam was acquired, with an identical readout to the diffusion-weighted scan, but with the phase encode direction flipped by 180° (in the anterior-posterior direction), for correction of susceptibility-related artefacts.

**MRI processing**. Structural parcellation was conducted using the T1 MPRAGE sequence processed using Connectome Mapper 3[48], which is written in Python and uses Nipype[49]. It is encapsulated in a BIDS app based in Docker and Singularity container technologies[50–52]. Resampling to isotropic resolution, the Desikan–Killiany brain parcellation[53] and brainstem parcellation[54] were applied using FreeSurfer 6.0.1[55]. Final parcellations were created by performing cortical brain parcellation at scale 3 of the Lausanne Atlas (v2018)[56], probabilistic atlas-based segmentation of the thalamic nuclei[34] and combination of all segmented structures, using in-house CMTK tools and the antsRegistrationSyNQuick tool of ANTS v2.2.0[57], to create 253 parcels for each subject (219 cortical parcels; 30 subcortical structures including 7 thalamic nuclei on each side; 4 brainstem structures including midbrain, pons, medulla and superior cerebellar peduncle).

Connectomes were generated in mrtrix using in-house scripts[58]. Following preprocessing (denoising, eddy correction and bias correction), response was estimated using multi-shell, multi-tissue constrained spherical deconvolution that exploits the unique $b$ value dependencies of different tissue types and produces more accurate apparent fibre density measures[59]. These diffusion images were rigidly registered to the structural images using reg_aladin[60], optimised using SIFT2[61] and probabilistic tractography using the iFOD2 algorithm in mrtrix[62] was performed using five million streamlines seeded from the entire white matter. Fractional anisotropy thresholds for terminating tracts were set to default values of 0.05 (0.1 × 0.5). This resulted in symmetric, weighted, undirected connectivity matrices.

All imaging was visually validated to verify the accuracy of the cortical segmentation and parcellation, ensure minimal movement artefact in the raw diffusion imaging and confirm accurate registration between the structural and diffusion images. No manual alterations were made to segmentation images. Following validation, three resective surgery patients and 1 control were excluded due to errors in the parcellation or diffusion processing—these were mostly related to diffusion processing errors that resulted in extreme occipital orientation distribution functions.

**Controllability metrics**. Connectivity matrices were analysed to ensure that the pre-requisites for controllability metrics were met, namely that no parcels or subnetworks were disconnected from the rest of the graph and that all diagonals were set to 0[14]. No thresholding was applied, in line with SIFT2 recommendations[61]. Average controllability and modal controllability were calculated for each parcel according to established methods[14,15]. Briefly, we employed a simplified noise-free linear discrete-time and time-invariant network model to describe the dynamics of the brain:

$$x(t + 1) = Ax(t) + B_\kappa u_\kappa(t), \qquad (1)$$

where $x : \mathbb{R}_{\geq 0} \to \mathbb{R}^N$ describes the state (i.e., activity level) of the parcels over time, and $A \in \mathbb{R}^{N \times N}$ is the structural connectivity matrix, which is subsequently stabilised by dividing by the mean edge weight. The input matrix $B_\kappa$ represents the control points $\kappa$, where $\kappa = \{k_1, \dots, k_m\}$ and $B_\kappa = [e_{k_1}, \dots, e_{k_m}]$ and $e_i$ denotes the i-th canonical vector of dimension N. The input $u_\kappa : \mathbb{R}_{\geq 0} \to \mathbb{R}^m$ describes the control strategy.

Average controllability of a network is equal to the average input energy from a set of control nodes and overall possible target states[63,64], which is proportional to the trace of the inverse of the controllability Gramian ---- $Trace(W_\kappa^{-1})$, where:

$$W_\kappa = \sum_{\tau=0}^{\infty} A^\tau B_\kappa B_\kappa^T A^{T\tau}. \qquad (2)$$

Regions with high average controllability tend to be densely connected, are involved in maintaining smooth and frequent transitions between related brain states and are enriched in the default mode network[15].

Modal controllability refers to the ability of a node to control each evolutionary mode of a dynamic network[65]. It is computed from the eigenvector matrix $V = [v_{ij}]$ of the structural connectivity matrix A. Specifically, modal controllability of the

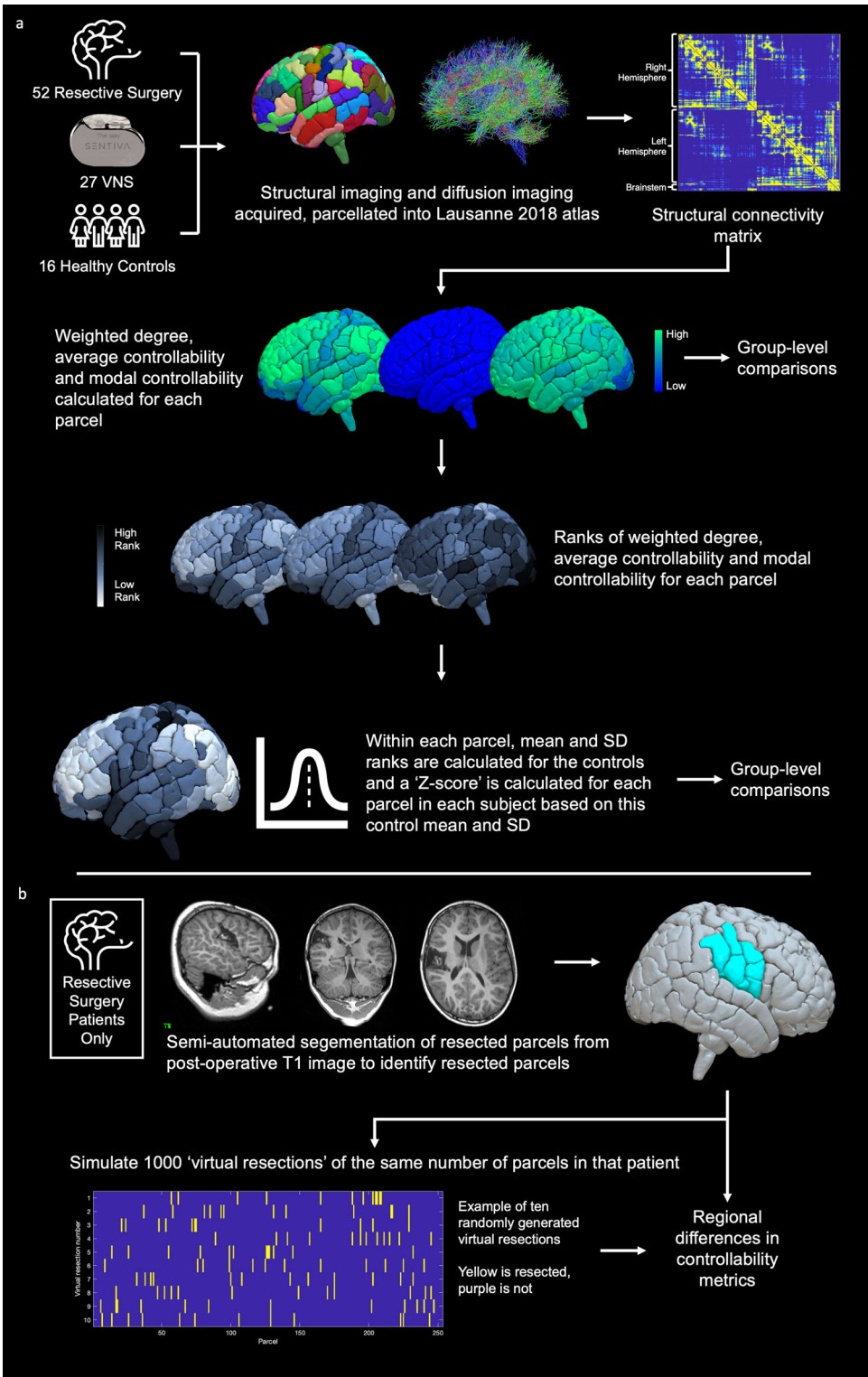

**Fig. 7 Summary of image processing workflow. a** Following subject identification, structural imaging was parcellated into Scale 3 of the Lausanne 2018 resulting in 253 parcels per patient. Structural connectivity matrices were generated for each patient where the connection strength was derived from the number of streamlines between each of the 253 parcels. Using these matrices, the weighted degree, average controllability and modal controllability were calculated for each parcel, which were subsequently ranked within each subject and normalised based on the mean and standard deviation of the ranks in the healthy control subjects. **b** For the resective surgery cohort, resection volumes were mapped to the pre-operative parcellation to identify resected parcels. In order to robustly assess any regional differences, 1000 virtual resections of the same number of parcels as the actual resection were also mapped for each patient.

parcel $i$ is defined as a scaled measure of the controllability of all N modes $\lambda_1(A), \ldots, \lambda_N(A)$ from itself:

$$\phi_i = \sum_{j=1}^{N}(1 - \lambda_j^2(A))v_{ij}^2. \qquad (3)$$

Regions of high modal controllability tend to be weakly connected areas, allow switching between brain states that have high energy barriers between them, and are enriched in frontoparietal and cingulo-opercular control systems[15].

In addition, we calculated the weighted degree of each node, defined as the sum of weights of edges connected to the node.

**Graph metrics**. All graph metrics were calculated using the Brain Connectivity Toolbox[66]. In order to probe the organisation of the graphs, metrics were chosen that were calculated on a whole graph basis and were assessed to reflect the level of organisation within the graphs.

*Density*. Fraction of present connections to possible connections

*Modularity*. The degree to which the network may be subdivided into clearly delineated groups according to the Louvain community detection algorithm[67].

*Global Efficiency*. Inverse of the mean shortest path length which is defined as the shortest distance between any pair of nodes. A measure of network efficiency that quantifies the ease of information transfer between two nodes; it requires knowledge of complete network structure[68].

*Diffusion efficiency*. Inverse of mean first passage time which is defined as the expected number of steps it takes a random walker to reach one node from another. An alternative measure of network efficiency that quantifies the ease of information transfer between nodes but when this information spreads by random diffusion; it is based on limited local knowledge of network topology[69].

**Statistics and reproducibility**. At the network level, raw values of mean weighted degree, mean average controllability, mean modal controllability, the correlation coefficients and graph metrics were compared, correcting for age, sex, cognitive function (full-scale IQ) and mean weighted degree using a GLM approach. The responses for mean modal controllability were not normally distributed (Shapiro–Wilk test) and so a gamma distribution was used for the GLM. Pairwise comparisons between the treatment groups were conducted using Fisher's least significant difference procedure for the GLMs. As a sensitivity analysis, two further GLM analyses were undertaken, one comparing the controls only the patients aged >12 and a second with the patients aged >12 and the covariates (age and full-scale IQ) removed. For all GLM analyses, we also calculated the effect size based on cohen's d using the estimated marginal means and standard deviations from the GLM; a value >0.5 can be interpreted as medium effect size and >0.8 as a large effect size.

For the node-level analyses, ranks were normalised to the control mean and standard deviation to result in a $z$ score for each parcel in each patient, which served to normalise the skewed distribution (Supplementary Fig. 1). Further analyses were conducted on these $z$ scores, comparing parcel-wise $z$ scores across groups. Multiple comparisons were corrected for by choosing a $z$ score threshold (±3.1) that adequately corrected for the number of comparisons (253).

For the resective surgery group, post-operative scans were segmented using semi-automated segmentation on ITK-SNAP[70]. Resection volumes were coregistered to the pre-operative scan parcellation to identify the parcels that had been resected (Fig. 7). Any overlap (e.g., partially resected) was considered a resected parcel for the purposes of analysis. Between 3 and 24 parcels (median 12) were resected in each patient and there was an expected representation of resections from most parts of the brain, with the most common resections involving the anterior and mesial temporal structures and right frontal lobe with less resections involving the posterior mesial and occipital structures (Supplementary Fig. 7). In addition, expected post-operative connectivity matrices were generated by re-running the tractography and excluding the resection volume using the mrtrix command tckexclude, as has been performed previously[71]. The mean z scores of each metric (weighted degree, average and modal controllability) in the resected parcels, non-resected parcels and post-operative connectivity matrices were compared between groups using two-tailed one or two-sample $t$ tests, whilst correcting for multiple comparisons using the Benjamini–Hochberg FDR method.

**Simulation modelling**. Modelling was conducted using the structural connectivity matrices of the healthy controls as a base to attempt to transform them to a matrix with similar network properties to that of the patients with epilepsy. Prior analyses were used to inform the parameters of the changes applied to them.

In the first modelling exercise, the overall graph weight was increased by adding edges (where there were none before) that followed the same distribution as the edge weights for all subjects (Weibull Distribution with $a = 0.717$ and $b = 0.325$; using the curve fitting toolbox in Matlab, Supplementary Fig. 3).

In the second modelling exercise, edge weights in the controls were increased by a low number of streamline counts, chosen at random from a distribution with mean of 50 and standard deviation of 15. This was informed by examining differences in the structural connectivity matrices between patients and controls (Supplementary Fig. 5). The number of edges was fixed at 1440/31818, accounting for 4.5% of all edges in the graph, which was informed by the mean weighted degree & density increase between controls and patients. For each control, 10 anatomically constrained models were generated, where the weight was added only to randomly chosen thalamocortical edges. In addition, for each control, 200 anatomically unconstrained null models were generated, where weight was added to randomly chosen ipsilateral connections. $z$ scores were generated for each controllability and graph metric within each subject from the distribution (mean and standard deviation) of the metrics in the null models and these were used to calculate the $z$ scores for metrics in the anatomically constrained models and baseline connectomes. $z$ scores were compared between groups using two-tailed two-sample $t$ tests and corrected for multiple comparisons using the Benjamini–Hochberg FDR method.

**Citation diversity statement**. Recent work in neuroscience has identified a bias in citation practices such that papers from women and other minority scholars are under-cited relative to the number of such papers in the field[72,73]. Here we sought to proactively consider choosing references that reflect the diversity of the field in thought, form of contribution, gender, race, ethnicity and other factors. First, we obtained the predicted gender of the first and last author of each reference by using databases that store the probability of a first name being carried by a woman. By this measure (and excluding self-citations to the first and last authors of our current paper), our references contain 10.63% woman(first)/woman(last), 21.61% man/woman, 13.45% woman/man and 54.31% man/man. A comparison to leading neuroscience journals is shown in Supplementary Fig. 8. This method is limited in that (a) names, pronouns and social media profiles used to construct the databases may not, in every case, be indicative of gender identity and (b) it cannot account for intersex, non-binary, or transgender people. Second, we obtained predicted racial/ethnic category of the first and last author of each reference by databases that store the probability of a first and last name being carried by an author of colour. By this measure (and excluding self-citations), our references contain 11.22% author of colour (first)/author of colour(last), 19.93% white author/author of colour, 23.53% author of colour/white author and 45.32% white author/white author. This method is limited in that (a) names and Florida Voter Data to make the predictions may not be indicative of racial/ethnic identity, and (b) it cannot account for Indigenous and mixed-race authors, or those who may face differential biases due to the ambiguous racialization or ethnicization of their names. We look forward to future work that could help us to better understand how to support equitable practices in science.

**Reporting summary**. Further information on research design is available in the Nature Research Reporting Summary linked to this article.

## Data availability

The individual patient structural connectivity matrices and accompanying demographic information are available on github (www.github.com/aswinchari/NetworkControl) and in Supplementary Data 1. The MRI scans are not available publicly due to potential patient identifiable information and confidentiality restrictions. Data for plotting Figs. 2–4 and 6 are available in Supplementary Data 2.

## Code availability

All statistical analyses were performed on Matlab v 2018b and SPSS v24. All code is available on github (www.github.com/aswinchari/NetworkControl) and the version used in this analysis has been DOI-minted[74]. The code, in conjunction with the connectivity matrices, allows for reconstruction of all figures in the manuscript; the 'Connectome Analysis' folder contains code for all the figures apart from Fig. 4, which is part of the 'Modelling' folder. Figures were made using Matlab and brain images were made using SurfIce, placed onto a parcellated paediatric average t1 brain (4.5–18.5 y)[75].

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

## Acknowledgements

A.C. is supported by a Great Ormond Street Hospital (GOSH) Children's Charity Surgeon Scientist Fellowship. This work has been supported by the GOSH-National Institute of Health Research Biomedical Research Centre.

## Author contributions

A.C., R.E.R. and R.C.S. conceived the study. A.C., K.S., X.H. and R.E.R. processed and analysed the data. All authors (including M.T., C.A.C. and D.S.B.) were involved in the interpretation of the data, drafting of the manuscript, and its subsequent editing. All authors approved the final version prior to submission.

## Competing interests

The authors declare no competing interests.
