## [Peer Review File · Communications Biology]

Reviewers' comments:

Reviewer #1 (Remarks to the Author):

The paper 'Drug-resistant focal epilepsy in children ...' by Chari et al. investigates brain connectivity in 16 healthy controls and 79 patients with different types of epilepsy. I will start noting that this is a topic of high interest to the field, that there is no lack of advanced analyses in this study, and the application of control theory to brain parcels is central here. The authors use a linear time invariant model to establish different flavors of controllability. In light of the quantity and quality of the analysis effort, I would anticipate a rich set of conclusions and I have to admit that I am rather disappointed with this aspect. The only conclusion I get away with is that there are differences across the patient groups, but I don't gain any clear insight. In addition, there is a finding with clinical relevance that allows one to identify resected parcels in one patient group. I do wonder though if this couldn't be accomplished as well by using simpler parameters (e.g., connectivity).

To avoid that this paper is just a compilation of data analyses and modeling, I also wonder if the authors could describe more concretely how the parameters are extracted from the data (i.e., an improved version of Fig. 7) and back, that is, how the results are interpreted in light of functional/pathological interactions across brain areas.

Detail: Should the last term in the Gramian, Equation (2), be $(AT)\tau$ rather than $(A)\tau$?

Wim van Drongelen
Professor
Section Pediatric Neurology
Computational Neuroscience
The University of Chicago

Reviewer #2 (Remarks to the Author):

In this study the authors generated structural connectomes from 79 children with epilepsy and 16 controls. The authors investigated network measures such as weighted degree and controllability. Authors found differences between these groups and interpreted this as patients having an ability to reach difficult to reach neurophysiological states (such as a seizure state).

I have some comments which I hope will help the authors to improve their interesting study.

My main concern with the study is that the patient and control groups are very different in terms of age. I recognise the authors attempted to correct for this using a linear model, but what evidence is there to suggest that a linear model is appropriate here? One way to convince me would be to repeat the analysis using only the subset of patients over 12 years old, and remove age from the GLM. Although some statistical power would be lost I would hope the effect sizes are similar (see later comment on effect size reporting). Similar for cognitive function (FSIQ), which I would suggest to remove from the GLM since it could be argued that reduced FSIQ is not independent of epilepsy pathology. Supplementary analyses here would be very welcome.

"Average controllability is used to measure the ability of a region to drive the brain to easy-to-reach states;" - Has this been demonstrated causally? Or is that the interpretation of this measure? Perhaps more subtle/cautions phrasing is necessary here.

"Existing reports of controllability in epilepsy are scarce" suggested additional citations:

<https://doi.org/10.3389/fnins.2015.00202>

<https://doi.org/10.3182/20140824-6-ZA-1003.00786>

I encourage the authors to consider their use of streamline count as a measure of connectivity in the context of the following article. Authors may wish to discuss the interpretation of streamline count as a measure of connectivity. <https://onlinelibrary.wiley.com/doi/full/10.1002/jmri.27188>

Streamline counts to larger areas will be inevitably higher. How was this issue handled / accounted for?

To generate post-operative matrices, the authors perform tractography on the pre-operative diffusion scan with the resection volume excluded. Thus, the authors are investigating the *expected* postoperative network, not the *actual* post-operative network since postoperative diffusion data are not used. This approach was first proposed in the following study which I would suggest to reference in this context: <https://doi.org/10.1016/j.nicl.2018.01.028>

Results: please report the effect size when reporting p values.

Generally, more detail is needed in the methods section "MRI processing", some examples below

- were eddy current and motion artefacts corrected? This is likely important given that these are scans of children with epilepsy. Did motion differ between groups?
- I had a look at the data on github and the connection matrices contain values that are not integers. How is it possible to have a fraction for an edge weight if the edge weight is number of streamlines?
- "applied using FreeSurfer 6.0.1." apparently the segmentation was checked. Did any corrections need to be made? surfaces checked for correctness? Please state more clearly.
- "Connectomes were generated in mrtrix" suggest to cite mrtrix
- "diffusion images were registered to the structural images," How? Rigid body? using ANTS?
- probabilistic tractography - please give further details on thresholds used for termination (angular threshold, FA etc)
- how were two regions defined as connected? If they had streamlines ending within them? passing through them? both?
- how many streamlines were discarded due to not meeting above criteria?

Minor:

line 530 - scam scan

line 697 show shown

Point-by-point Responses to Reviewers' Comments

Reviewer #1

The paper 'Drug-resistant focal epilepsy in children ...' by Chari et al. investigates brain connectivity in 16 healthy controls and 79 patients with different types of epilepsy. I will start noting that this is a topic of high interest to the field, that there is no lack of advanced analyses in this study, and the application of control theory to brain parcels is central here. The authors use a linear time invariant model to establish different flavours of controllability. In light of the quantity and quality of the analysis effort, I would anticipate a rich set of conclusions and I have to admit that I am rather disappointed with this aspect. The only conclusion I get away with is that there are differences across the patient groups, but I don't gain any clear insight. In addition, there is a finding with clinical relevance that allows one to identify resected parcels in one patient group. I do wonder though if this couldn't be accomplished as well by using simpler parameters (e.g., connectivity).

Thank you for the encouraging comments. The rationale to use this approach was not to identify resectable parcels but to use controllability to understand something about the functional dynamics from the structural connectivity matrix.

We have added to the introduction and discussion to shed more light on the implications of these findings and the importance of integrating structural and functional network data under the same framework.

Introduction:

In testing and validating these hypotheses, our study adds weight to the concept of network-focused interventions in drug-resistant epilepsy. By linking network topology and dynamics, controllability allows us to link the pathophysiology of epilepsy and the possible treatments in one unified framework. In this study, the framework has been applied to surgical treatments which modify the network topology but potentially applies to neuromodulatory treatments too, in which the network topology is unaltered but is harnessed to guide optimal location and stimulation parameters. Ultimately, controllability is attractive as it allows the fusion of multimodal structural and functional network data under the same framework.

Discussion:

At whole brain scale, we have progressed our understanding of brain network dysfunction in drug-resistant epilepsy, showing that the structural network alterations are constrained to more easily reach nearby and distant neurophysiological states given a perturbation. This perturbation may be caused by extrinsic inputs or intrinsic local abnormalities that, for example, affect excitatory-inhibitory balances, leading to a transition into a seizure state. The graph metrics and correlations illustrate that this is a function of a less organised and less modular system, which the simulations suggest may in turn be due to increased thalamocortical connectivity.

Other studies have used the control theory framework to simulate responses of the brain to external perturbation via stimulation. At node level, our demonstration of regional differences in modal controllability in children with drug-resistant epilepsy raises the potential to use these metrics to guide clinical treatments. Firstly, there is the possibility of identifying parcels of abnormal controllability as surrogate markers of the epileptogenic zone. Such analyses may reveal more focal, more widespread or multifocal network abnormalities that could be useful in guiding resections and prognosticating. Secondly, the network dynamics from the control theory framework could be used to choose the location and stimulation parameters for neuromodulation treatments, with the aim of normalising the network dynamics; this has the potential to ameliorate both the susceptibility to seizures and some of the comorbid conditions such as cognitive and behavioural dysfunction associated with epilepsy.

To avoid that this paper is just a compilation of data analyses and modelling, I also wonder if the authors could

describe more concretely how the parameters are extracted from the data (i.e., an improved version of Fig. 7) and back, that is, how the results are interpreted in light of functional/pathological interactions across brain areas.

Thank you. As per your and Reviewer #2's comments, we have expanded on the methods. We feel that the methods summary figure adequately captures the methods used and have expanded on the interpretation as per the point above.

Detail: Should the last term in the Gramian, Equation (2), be (AT)tau rather than (A)tau?

Thank you. Amended

Reviewer #2 (Remarks to the Author):

In this study the authors generated structural connectomes from 79 children with epilepsy and 16 controls. The authors investigated network measures such as weighted degree and controllability. Authors found differences between these groups and interpreted this as patients having an ability to reach difficult to reach neurophysiological states (such as a seizure state).

I have some comments which I hope will help the authors to improve their interesting study.

My main concern with the study is that the patient and control groups are very different in terms of age. I recognise the authors attempted to correct for this using a linear model, but what evidence is there to suggest that a linear model is appropriate here? One way to convince me would be to repeat the analysis using only the subset of patients over 12 years old, and remove age from the GLM. Although some statistical power would be lost I would hope the effect sizes are similar (see later comment on effect size reporting). Similar for cognitive function (FSIQ), which I would suggest to remove from the GLM since it could be argued that reduced FSIQ is not independent of epilepsy pathology. Supplementary analyses here would be very welcome.

Thank you. In the first section of the results, we have performed sensitivity GLM analyses (i) removing age, sex and cognition for all 95 patients and (ii) removing age, sex and cognition for only the 44 patients aged >12 in all 3 groups. The results are included as Supplementary Results 1. Both analyses show largely consistent findings, with little difference in weighted degree and average controllability metrics but significant differences in modal controllability between children with drug-resistant epilepsy and controls:

Supplementary results 1: Sensitivity analysis for group-level comparisons

(i) Repeated analysis without adjusting for age, sex and cognitive function in all 95 patients:

- **Weighted Degree: No significant differences between groups, not consistent with corrected findings (control vs resective surgery cohen's $d = 0.2$, $p = 0.58$, control vs VNS cohen's $d = 0.1$, $p = 0.85$ and resective surgery vs VNS cohen's $d = 0.1$, $p = 0.68$).**
- **Average Controllability: Significant differences between groups, consistent with corrected findings (control vs resective surgery cohen's $d = 0.33$, $p = 0.25$, control vs VNS cohen's $d = 1.1$, $p = 3 \times 10^{-4}$ and resective surgery vs VNS cohen's $d = 0.80$, $p = 0.001$).**
- **Modal Controllability: Significant differences between groups, consistent with corrected findings (control vs resective surgery cohen's $d = 1.3$, $p = 4 \times 10^{-6}$, control vs VNS cohen's $d = 2.0$, $p = 5 \times 10^{-11}$ and resective surgery vs VNS cohen's $d = 0.8$, $p = 9 \times 10^{-4}$).**

(ii) Repeated analysis without adjusting for age, sex and cognitive function in only 44 patients aged >12:

- **Weighted Degree: Significant differences between groups, partially consistent with corrected findings (control vs resective surgery cohen's $d = 0.85$, $p = 0.14$, control vs VNS cohen's $d = 0.4$, $p = 0.30$ and resective surgery vs VNS cohen's $d = 0.4$, $p = 0.25$).**

- **Average Controllability: Significant differences between groups, partially consistent with corrected findings (control vs resective surgery cohen's $d = 0.3$, $p = 0.48$, control vs VNS cohen's $d = 0.8$, $p = 0.03$ and resective surgery vs VNS cohen's $d = 0.6$, $p = 0.14$).**
- **Modal Controllability: Significant differences between groups, consistent with corrected findings (control vs resective surgery cohen's $d = 1.0$, $p = 0.005$, control vs VNS cohen's $d = 1.4$, $p = 2 \times 10^{-4}$ and resective surgery vs VNS cohen's $d = 0.4$, $p = 0.30$).**

"Average controllability is used to measure the ability of a region to drive the brain to easy-to-reach states;" - Has this been demonstrated causally? Or is that the interpretation of this measure? Perhaps more subtle/cautions phrasing is necessary here.

Thanks. This is an important point and we have clarified this in the manuscript when we introduce the measures. We have stressed that this is a theoretical framework but has since been applied to real world data in the introduction:

Originally derived from theoretical descriptions of linear dynamics in networks, the definitions of average and modal controllability have since been successfully applied to model responses to electrical stimulation in practice.

"Existing reports of controllability in epilepsy are scarce" suggested additional citations:

<https://doi.org/10.3389/fnins.2015.00202>

<https://doi.org/10.3182/20140824-6-ZA-1003.00786>

Thanks. We have included these citations.

I encourage the authors to consider their use of streamline count as a measure of connectivity in the context of the following article. Authors may wish to discuss the interpretation of streamline count as a measure of connectivity. <https://onlinelibrary.wiley.com/doi/full/10.1002/jmri.27188>

Thanks. We have used state-of-the art methods that account for many of the limitations mentioned in the above paper. Specifically, we use anatomically constrained tractography from multi-shell multi-tissue diffusion imaging and apply spherical-deconvolution informed filtering of tractograms (SIFT2) that the streamline densities generated are as close as possible to the fibre densities estimated by constrained spherical deconvolution. We have added references to this in the methods:

Connectomes were generated in mrtrix using in-house scripts.1 Following preprocessing (denoising, eddy correction and bias correction), response was estimated using multi-shell, multi-tissue constrained spherical deconvolution that exploits the unique b-value dependencies of different tissue types and produces more accurate apparent fibre density measures.1 These diffusion images were rigidly registered to the structural images using reg_aladin1, optimized using SIFT21 and probabilistic tractography using the iFOD2 algorithm in mrtrix1 was performed using 5 million streamlines seeded from the entire white matter. Fractional anisotropy thresholds for terminating tracts were set to default values of 0.05 (0.1 x 0.5). This resulted in symmetric, weighted, undirected connectivity matrices.

Streamline counts to larger areas will be inevitably higher. How was this issue handled / accounted for?

This was not accounted for, although in the node-level analyses, differences in rank of weighted degree, average and modal controllability were considered.

To generate post-operative matrices, the authors perform tractography on the pre-operative diffusion scan with the resection volume excluded. Thus, the authors are investigating the *expected* postoperative network, not the *actual* post-operative network since postoperative diffusion data are not used. This approach was first proposed in

the following study which I would suggest to reference in this context: <https://doi.org/10.1016/j.nicl.2018.01.028>

Thank you – we have referenced this in the methods.

Results: please report the effect size when reporting p values.

Thank you – we have reported effect sizes for all the main mean and parcel-wise analyses. For the GLM models, we also used the estimated marginal means and standard deviations to report a cohen's d statistic as a measure of effect size.

Generally, more detail is needed in the methods section "MRI processing", some examples below

- were eddy current and motion artefacts corrected? This is likely important given that these are scans of children with epilepsy. Did motion differ between groups?

We did not have a readout of motion, but the scans were visually checked for motion artefacts and scans with significant artefact were excluded. The vast majority were acquired under general anaesthesia, so motion was not an issue. The diffusion pre-processing involved eddy correction and we have added this.

- I had a look at the data on github and the connection matrices contain values that are not integers. How is it possible to have a fraction for an edge weight if the edge weight is number of streamlines?

This is because the tcksift2 command was used to 'optimise per-streamline cross-section multipliers to match a whole-brain tractogram to fixel-wise fibre densities'. We have referenced this.

- "applied using FreeSurfer 6.0.1." apparently the segmentation was checked. Did any corrections need to be made? surfaces checked for correctness? Please state more clearly.

No manual corrections were made and we have added this.

- "Connectomes were generated in mrtrix" suggest to cite mrtrix

Cited

- "diffusion images were registered to the structural images," How? Rigid body? using ANTS?

We have expanded.

- probabilistic tractography - please give further details on thresholds used for termination (angular threshold, FA etc)

We have added these details – they were the default values for the iFOD2 algorithm on mrtrix.

- how were two regions defined as connected? If they had streamlines ending within them? passing through them? both? how many streamlines were discarded due to not meeting above criteria?

In line with the guidance for tcksift2, thresholding using a minimum number of streamlines threshold was not applied. We have added this and the relevant reference

REVIEWERS' COMMENTS:

Reviewer #2 (Remarks to the Author):

The authors have addressed all my points and I congratulate them on a nice piece of work.